# Comparing High Entropy Reinforcement Learning to Navmesh for Automated Collision Bug Testing

**Aurélien Chambon[1], Nicolas Grelier[1]**

`{aurelien.chambon, nicolas.grelier}@pullupent.com`

[1]**PulluP Entertainment**

## Abstract

Automatic bug testing in video games is a topic of growing interest for industry and academia. Reinforcement learning (RL) is emerging as a promising approach, particularly for detecting collision bugs, which are tedious and time-consuming for Quality Assurance (QA) analysts to evaluate manually. However, prior works often rely on visual inspection of results of RL navigation agents. In this short paper, we introduce an early-stage method for automating collision bug detection by comparing the traversal time of an RL navigation agent with that of a navmesh agent. We pretrain this agent in an obstacle free environment, deploy it along a route and exploit entropy-driven RL exploration to bypass obstacles, before resetting to the pretrained policy to continue map coverage. This approach enables scalable map analysis for detecting clipping and collision issues, while automatically flagging anomalies for developer review. Additionally, we provide early insights on the impact of high entropy tuning in our method.

## 1   Introduction

The video game industry has grown steadily over the past few decades. Budgets for major titles now reach hundreds of millions of dollars, enabling greater complexity in gameplay, graphics, and map size. Like all software, video games are prone to bugs, which can disrupt immersion or force players to restart levels. To minimize such issues in commercial releases, developers and publishers perform QA testing. This often involves identifying collision bugs that let players bypass obstacles or get stuck where they should move freely.

To our knowledge, these tests are typically tedious and time-consuming, requiring exhaustive checks across all obstacles, directions, and platform builds (e.g., PC, Xbox, PlayStation, Nintendo). Existing research on automatic collision bug detection using RL has shown promise; however, most prior work relies heavily on visual inspection to identify bugs and does not scale to full-map evaluations.

In this short paper, we present an early idea for collision tests automation using a high entropy RL agent. To detect missing collisions and stuck states, we compare the agent's traversal time to that of the navmesh: a precomputed shortest path between any two reachable points in the level. This allows to automatically notifying developers for further review. We also present early insights on the impact of high entropy tuning in our method.

## 2   Related Work

The application of RL in video games initially focused on using games as controlled environments for evaluating and benchmarking algorithms. This approach can be referred to as *games for AI* (Yannakakis & Togelius, 2018). More recently, the demonstrated success of RL agents achieving superhuman performance in these environments has spurred the integration of adversarial RL-

based agents into commercial games, such as *Gran Turismo* (Wurman et al., 2022) or *For Honor* (Bairamian et al., 2024), marking the approach of *AI for games*.

In the context of AI for games, RL-based agents are employed not only to play alongside or against human players, but also for Quality Assurance (QA) testing. These automated tests fall into two main categories: *gameplay testing* (difficulty evaluation, player behavior modeling), and *technical testing* (collision detection bugs and frame rate anomalies) (Le Pelletier de Woillemont et al., 2022).

In the literature, gameplay testing is often approached through Imitation Learning (IL) to replicate human-like behavior when evaluating elements that are hard to measure with standard metrics (Ahlberg et al., 2023). While gameplay testing often benefits from the expertise of QA analysts and designers to iteratively refine game mechanics and user experience, technical testing typically involves repetitive tasks better suited for automation, like map walkthroughs. Gillberg et al. (2023) leveraged Proximal Policy Optimization (PPO) to propose an approach replacing traditional scripted solutions for main path QA testing where scripting becomes impractical due to the high degree of navigational freedom in *Dead Space (2023)* and *Battlefield 2042*.

Beyond main path evaluation, QA testing must also ensure the absence of technical issues such as collision bugs, ensuring obstacles cannot be passed through. Bergdahl et al. (2020) used PPO to detect collision-related bugs by tasking randomly initialized agents with locating randomly placed orbs, visually exposing missing collision boundaries. However, their approach requires manual inspection and is limited to individual obstacles. In contrast, Gordillo et al. (2021) incorporated intrinsic curiosity into the reward function to promote broad exploration across the entire map, using terminal states to identify stuck regions. Like the former, they rely on visual 3D path renderings to detect collision issues. In this early work, we focus on *AI for games* applied to *technical QA testing*. We aim to extend these two approaches by developing a method relying on high entropy RL for systematically investigating entire game maps to detect collision bugs and efficiently notify developers of potential issues.

## 3 Proposition

In this section, we introduce our method for collision bug finding based on RL. The core idea is to compare the time taken by the RL agent to reach target locations against that of a navmesh agent, which serves as ground truth. The pipeline, illustrated in Figure 1, consists of two components: route generation and walkthrough, along with collision checking and a reset strategy.

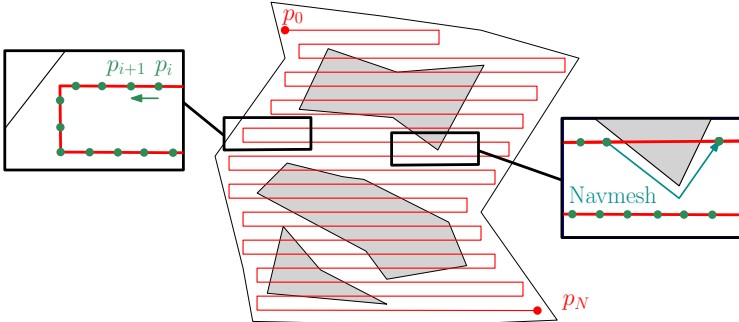

Figure 1: Illustration of the proposed method

### 3.1 RL setup and Overview of the method

In our approach, a navigation RL agent is used to reach a target, following a policy $\pi$ using a policy gradient method based on entropy regulation, within a Partially Observable Markov Decision Process parametrized by the tuple $(\mathcal{S}, \mathcal{A}, \mathcal{T}, R, \Omega, \mathcal{O}, \gamma, H, N_{\max})$. Here, $\mathcal{S}$ denotes the state space, encompassing all information available within the game engine that characterizes the agent's

state. The observable portion of this space is $\mathcal{O}$, comprising the agent's horizontal velocity $(\mathbb{R}^2)$ and forward vector angle $(\mathbb{R})$ both relative to the target, the target's position relative $(\mathbb{R}^2)$ to the agent, and nine raycasts proportions $([0,1]^9)$. Each raycast is sent within the agent frame of reference, originates from $[0,0,h]$ and extends $l_{\text{ray}}$ meters toward $[\cos(\theta_i), \sin(\theta_i), h]$ with $\theta_0 = 0$ and $\forall 1 \le i \le 4, \theta_i = \frac{\pi}{4i}, \theta_{i+4} = -\theta_i$. The parameter $h$ corresponds to half of the agent's capsule collider height. The action space is defined as $\mathcal{A} = [-1,1]^2$, including two continuous actions for forward/backward movement and rotation. $\mathcal{T}$ is the transition function, $R : S \times [0,H] \to \mathbb{R}$ denotes the reward function that is bound to change, with $H$ being the episode horizon in timesteps. We use a fixed discount factor $\gamma = 0.99$, and introduce $N_{\max}$ as the configurable maximum amount of steps allowed in the environment for a given task.

During pretraining, the agent operates in an obstacle-free environment of size $A \times A \, \mathrm{m}^2$ and is tasked with reaching a randomly placed target point. Once the target is reached, it is reset to a new random location. The agent is trained using a specific reward function $R_{\text{Pr}}$, designed to incentivize rapid target reaching. For $s \in \mathcal{S}$ and $t \in [0,H]$, let $\Delta_{\text{Tar}}(s,t)$, or simply $\Delta_{\text{Tar}}$, denote the Euclidian distance between the target and the agent at timestep $t$. The reward $R_{\text{Pr}}$ consists in a weighted sum of a flat reward for reaching the target, a continuous reward representing the speed towards the target, and a third component that decreases with each timestep.

$$R_{\text{Pr}}(s,t) = w_1 \mathbb{1}_{\{0\}}(\Delta_{\text{Tar}}) - w_2 \frac{d\Delta_{\text{Tar}}}{dt} - w_3 t \text{ with } (w_1, w_2, w_3) \in \mathbb{R}_+^3 \tag{1}$$

Once the agent has learned to navigate efficiently, the resulting policy $\pi_{\text{Pr}}$ is stored. We leverage this pretrained behavior to guide the agent along a predefined route that covers the entire map, divided into small segments. This allows us to detect if the agent can get stuck or clip through the environment.

### 3.2 Route generation and walkthrough

First, we manually split the map into connected subareas, defined as regions that are reachable without jumping or interacting with in-game elements. Within each subarea, we build a grid-based route designed to provide full coverage of each accessible region within the level (Figure 1).

For each route, the agent is initialized at a designated starting point $p_0$ and follows the predefined route, which is divided into small segments of fixed length $l_s$, until it reaches the endpoint $p_N$. Upon completing a route in one area, the agent is teleported to the starting point of the next area.

For each segment $[p_i, p_{i+1}]$, we record the time (in timesteps) taken by the RL agent to travel the segment $\delta_i^{\text{RL}}$ and compare it to the reference time $\delta_i^{\text{Nav}}$ of a navmesh agent, serving as ground truth. If the RL agent takes significantly longer than the navmesh $\left(\delta_i^{\text{RL}} > \delta_i^{\text{Nav}}\right)$, or fails to reach point $p_{i+1}$, the segment is flagged as anomalous and reported to the development team. Such anomalies may indicate areas where players can fall through the map or become stuck.

### 3.3 Collision checking and reset strategy

Obstacles can be readily detected when $p_{i+1}$ lies within a mesh. In such cases, the agent initiates a RL phase to learn how to reach the next reachable point as fast as possible. The reward function is changed to $R_{\text{Obs}}$, to encourage the agent to reach the target location in minimal time while avoiding the obstacle. Let $\Delta_{\text{Obs}}$ denote the distance between the nearest obstacle and the agent at timestep $t$.

$$R_{\text{Obs}}(s,t) = w_1 \mathbb{1}_{\{0\}}(\Delta_{\text{Tar}}) - w_2 \frac{d\Delta_{\text{Tar}}}{dt} - w_3 t - w_4 \mathbb{1}_{\{0\}}(\Delta_{\text{Obs}}) \text{ with } w_4 \in \mathbb{R}_+ \tag{2}$$

Initially, the agent attempts to reach the target by running directly into the obstacle, which is what we desire in order to uncover missing collisions. Over time, its entropy-driven exploration enables it to learn avoidance using the raycast observations. In parallel, we compute $\delta_i^{\text{Nav}}$ and use it to implement

an adaptive reset and bypass strategy. Specifically, for each segment, the episode horizon $H$ and the maximum duration of the navigation task $N_{\max}$ are set proportionally to $\delta_i^{\mathrm{Nav}}$. This prevents the agent from exploring irrelevant distant regions or being stuck on the task.

If the RL agent successfully reaches the target, we compare its best completion time $\delta_i^{\mathrm{RL}}$ against $\delta_i^{\mathrm{Nav}}$. If the RL agent is faster $\left(\delta_i^{\mathrm{Nav}} > \delta_i^{\mathrm{RL}}\right)$, this may indicate a collision issue such as passing through geometry or a flaw in the navmesh representation. Conversely, if the RL agent consistently fails to reach the target within the allocated time, it is a warning to investigate if the navmesh agent's success is intended. Both are flagged and reported to the development team. Finally, the policy is reset back to $\pi_{\mathrm{Pr}}$ once the obstacle has been successfully bypassed, allowing the agent to return to its original exploratory behavior.

## 4 Early Insights: High Entropy Weight Tuning

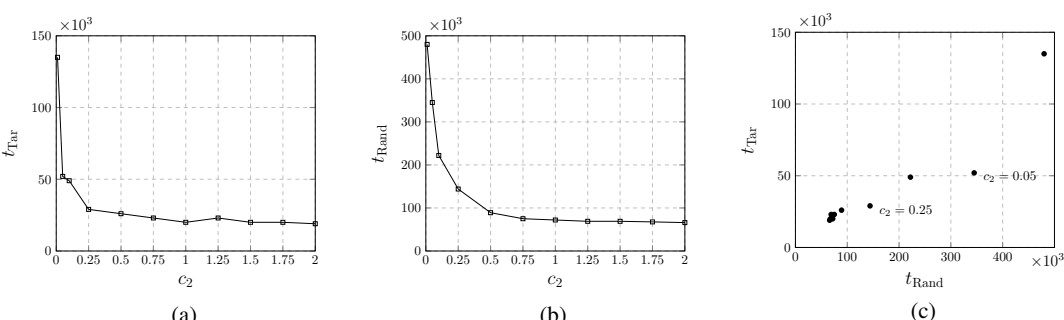

Figure 2: Entropy tuning analysis on time to reach target and reaching a random behavior

We use a modified Learning Agents plugin for a 3D game on Unreal Engine (details on modifications and hyperparameters are provided in the Appendix). It is based on a recurrent PPO, and therefore incorporates a weighted entropy bonus in the loss function Schulman et al. (2017):
$$L_t^{CLIP+VF+S}(\theta) = \hat{\mathbb{E}}_t[L_t^{CLIP}(\theta) - c_1 L_t^{VF}(\theta) + c_2 S[\pi_\theta](s_t)].$$

In this section, we study how the entropy coefficient $c_2$ affects the agent's ability to diverge from $\pi_{\mathrm{Pr}}$ to bypass obstacles and find collision bugs. We hypothesize higher entropy improves exploration but excessive values lead to erratic behavior, easily detected as near immobility due to random rotation.

Post-pretraining, we deploy the agent in a small environment with an obstacle and a target behind it. In Figure 2, we evaluate different $c_2$ values by (a) measuring the timestep to reach the target $t_{\mathrm{Tar}}$, (b) the timestep to reach a random behavior $t_{\mathrm{Rand}}$, and (c) the relation between the two. Our early insights show that decreasing the number of timesteps needed to reach the target comes at the price of decreasing the number of timesteps before the agent acts erratically. As we want to minize the former, and maximize the latter, one has a choice to make.

## 5 Conclusion, limitations and future work

In this short paper, we presented the early idea of a method for automating collision testing across an entire map by comparing the travel time of a RL-based navigation and a navmesh agent. Our approach allows for automatic flagging of anomalies, which are then notified to the developers.

Two key limitations are reliance on the navmesh as ground truth; inaccurate navmesh baking undermines our method's validity. Secondly, the agent's inability to jump limits jump-related bug findings, as remaining grounded is necessary for valid travel time comparisons with the navmesh agent.

In future work, we aim to empirically validate the method, examine the necessity of resetting to the pretrained policy for effective bug detection, incorporate additional actions, and further analyze the tradeoff related to the entropy coefficient and its impact on exploration and bug discovery.

## Appendix

In this appendix, we detail the modifications made to the plugin Learning Agents for Unreal Engine and the parameters used for the experiment described in section 4.

Notably, we revert to the theoretical PPO from Schulman et al. (2017). First, the advantage is normalized rather than clipped. Additionally, we apply an L2 loss for the critic and implement the PPO clipped loss function that incorporates the advantage term within the $\min$ operation:

$$L_t^{CLIP}(\theta) = \min(r_t(\theta)\hat{A}_t, \text{clip}(r_t(\theta), 1 - \epsilon, 1 + \epsilon)\hat{A}_t) \tag{3}$$

For reproducibility, the full set of parameters used in section 4 is listed in Table 1. These parameters were obtained empirically, by manual testing and fine-tuning.

Table 1: Parameters used in section 4

| VARIABLE | VALUE |
|---|---|
| Agents in parallel | 4 |
| Rollout buffer size | $1e4$ |
| Learning rate Policy & Critic | $1e-4$ |
| Policy & Critic Batch size | 512 |
| Policy window size | 8 |
| Critic warmup iterations | 8 |
| Iterations per gather | 4 |
| Epsilon clip | 0.2 |
| GAE Lambda | 0.95 |
| Discount factor | 0.99 |
| Policy & Critic Neural Network size | $1 \times 128$ |
| Memory state size | 64 |
| $c_1$ | $1e-3$ |
| $A$ | 35m |
| $w_1$ | 20 |
| $w_2$ | $1e-2$ |
| $w_3$ | $1e-3$ |
| $w_4$ | 50 |
| $l_s, l_{\text{ray}}$ | 20m |

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
