# OpenReview forum: "Comparing High Entropy Reinforcement Learning to Navmesh for Automated Collision Bug Testing"
_rl-conference.cc/RLC/2025/Workshop/RLVG — RLVG Workshop - RLC 2025_

### Official Review · Reviewer_4o48 · 2025-06-15
**The method is novel, seems to have merit and is described well; however, the evaluation of the method should be improved upon.**

**Rating:** 3
**Confidence:** 4

**Summary:**

The authors proposed a novel method of automated collision bug testing which utilises both navmesh and reinforcement learning. They suggest that by comparing the travel time between points $p_i$ and $p_{i+1}$ of an agent navigating with navmesh and another agent navigating with a high entropy reinforcement learning policy, the difference in traversal time $\delta^{\text{Nav}}_i >\delta^{\text{RL}}_i$ can be used as an indicator for missing collision boxes or a faulty navmesh bake.

**Strengths:**

Overall, the paper is well written, and the method is explained and motivated succinctly. Intuitively, the proposed method seems to have promise. Navmeshes have been used in gaming for a long time and are well understood, while RL is more dynamic and can both explore and exploit a game environment. Combining these two methods could compliment each other. The method is also, to my knowledge, a novel idea.

**Weaknesses:**

The work is missing key results which would validate their method. Although an early investigation, it is hard to be confident in their claim that their method “enables scalable map analysis for detecting clipping and collision issues, while also automatically flagging anomalies for developer review” without showcasing any evaluation to back their claim. Additionally, while the analysis of different entropy weights is interesting, it is seems like a minor detail in the overall method. It would be much more interesting to see an early stage application, or even toy example, of the proposed method.

**Best Paper Nomination:**

No

**Claims:**

The authors claim that their proposed method of automated collision bug testing is a novel one which would allow for “…scalable map analysis for detecting clipping and collision issues, while automatically flagging anomalies for developer review.”. This is in contrast to other RL methods for collision bug detection which require manual inspection after the running the agent.

**Suggestions:**

Firstly, it would be in the authors’ best interest for this workshop to set up a toy example of the proposed method in action. Create something similar to Figure 1 and artificially remove and add collision boxes. Some metrics of interest would be the average travel time of the navmesh and RL agent, the ratio of bugs detected. It would also be good to showcase how the method can be used to *automatically* highlight anomalies. While not directly applicable, look into how bugs are automatically highlighted in *Automated gameplay testing and validation with curiosity-conditioned proximal trajectories* by Sestini et al. They flag trajectories with intrinsic reward over a threshold as suspicious. Maybe the difference in travel time over a certain threshold can be used and highlighted in a similar manner?

Secondly, in the introduction of the paper, claims of the current state of QA testing within the gaming industry are made without any sources or justification. It would be in the authors’ interest to add some references with similar claims.

---

### Official Review · Reviewer_o3EK · 2025-06-16
**Detecting Collision Bugs in Games using Reinforcement Learning**

**Rating:** 3
**Confidence:** 2

**Summary:**

In this paper, the authors propose a method for detecting collision bugs using reinforcement learning. To do this, they propose comparing nav-mesh approaches with a reinforcement learning agent to see if one takes significantly more time than the other. They propose this as a way to help out with automated QA in games, which is an important and active research area. Their method involves utilizing entropy in conjunction with a PPO algorithm.

**Strengths:**

#1: The proposed method is quite interesting as it extends PPO and does some hyperparameter tuning on the entropy coefficient.
#2: The problem they're trying to solve is important to the field of reinforcement learning and automated quality assurance as a whole.

**Weaknesses:**

#1: The choice of hyperparameters is unclear to me regarding how they were selected.
#2: The details of the domain are also not precise.
#3: There are several dropped equations in the text without explaining the equation's relevance or significance.
#4: Initial empirical results just varied the entropy coefficient.
#5: Several other details (see suggestions below).

**Best Paper Nomination:**

No

**Claims:**

The authors provide some evidence to support their claims, but they primarily present plots illustrating how they adjust the value of the entropy coefficient. It would be helpful if the authors included more information on the domain they're considering, as well as how they set the hyperparameters to the chosen values.

**Suggestions:**

#1: The equation for the reinforcement learning agent taking longer than the navmesh would occur if it's 1 time step more. I think this should be a weighted equation, such as delta_i^(RL)>c*delta_i^(Nav).
#2: Between lines 85 and 86, there's a dropped equation that isn't explained in the text. It's always a good idea to explain the equations in a paper. In particular, it'd also be helpful to explain how the hyperparameters w_1, w_2, and w_3 are chosen.
#3: Provide a more detailed explanation of the domain. Is it just 2d games? I was a bit confused in Section 3.1 when it was discussed that several of the terms in the setup were in R^2.
#4: Explain why collision bugs, specifically within QA, are interesting to look at.
#5: LaTeX on line 72, such as using \cos instead of $cos$.
#6: Unclear diagram in 4 part (c), it would be helpful if the x's are replaced with dots.
#7: The end of Section 4 could be stronger. In particular, the line "As we want to minimize the former, and maximize the latter, one has a choice to make." How would you make this choice? You can mention this is future work, but there are many tradeoffs in machine learning, and solving them is important here.
#8: Line 136 to 137: I think it should be "the ability to notify developers".
#9: Line 138, it's unclear what you mean by "inaccurate navmesh baking"

---

### Decision · Program_Chairs · 2025-06-19

**Decision:**

Accept

**Comment:**

This paper presents an RL-based collision bug detection method in video games by comparing the traversal time of an RL agent with a nav-mesh agent to detect anomalies like clipping or missing collisions. The work presents a novel solution for a common task in video game development and testing. As pointed out by the reviewer, some details required for reproducibility and a clear explanation of some equations and diagrams are missing. We strongly encourage the authors to address these points in the camera-ready version.